# Intestinal Permeability and Depression in Patients with Inflammatory Bowel Disease

**DOI:** 10.3390/jcm11175121

**Published:** 2022-08-30

**Authors:** Miorita Melina Iordache, Cristina Tocia, Mariana Aschie, Andrei Dumitru, Mihaela Manea, Georgeta Camelia Cozaru, Lucian Petcu, Sabina E. Vlad, Eugen Dumitru, Anca Chisoi

**Affiliations:** 1Faculty of Medicine, Ovidius University of Constanta, 1 Universitatii Alley, 900470 Constanta, Romania; 2Prof. Alexandru Obregia Psychiatry Hospital, 10 Berceni Str., 041914 Bucharest, Romania; 3“St. Apostol Andrew” Emergency County Hospital, 145 Tomis Blvd., 900591 Constanta, Romania; 4Research Center for the Morphological and Genetic Study in Malignant Pathology (CEDMOG), Ovidius University of Constanța, 145 Tomis Avenue, 900591 Constanta, Romania; 5Academy of Romanian Scientists, 3 Ilfov Street, 050045 Bucharest, Romania; 6Faculty of Dental Medicine, Ovidius University of Constanta, 1 Universitatii Alley, 900470 Constanta, Romania

**Keywords:** leaky gut syndrome, biomarkers, psychiatric disorders, calprotectin, zonulin, LBP, IFAB

## Abstract

Depression is a global health problem that requires an early and accurate diagnosis to ensure quick access to appropriate treatment. Among multiple psychopathological paths, recent attention has focused on analysing the brain–gut–microbiota axis. The intestinal barrier plays a key role, and dysfunctions occurring at this level have implications for mental health. The aim of the present study was to investigate the role of intestinal permeability biomarkers, i.e., calprotectin, zonulin, lipopolysaccharide-binding protein (LBP) and intestinal fatty acid-binding protein (I-FAB), in relation to depression in patients with inflammatory bowel disease (IBD). This is the first study of this kind taking place in Romania, Eastern Europe, with an emphasis on patients with Crohn’s disease and ulcerative colitis. The correlations identified between depression and calprotectin and depression and LBP have the potential to shed light on the process of rapid diagnosis of depression with the help of biomarkers. Since depression is correlated with a decrease in the quality of life in patients with IBD, the need for access to appropriate treatments must be urgent.

## 1. Introduction

Depression is an individual, professional and social disability associated with somatic morbidities [1,2] and with a high risk of mortality from suicide [3]. However, access to specialists, a correct diagnosis and an effective cure are limited [4]. An early and accurate diagnosis [5,6], as well as a personalised selection of treatment [7], are necessary for diminishing the suffering and financial burden of the disease [3].

According to the diagnostic criteria in the Diagnostic and Statistical Manual of Mental Disorders (DSM V) [8], depression is an affective disorder with a heterogeneous group of symptoms. In order to be diagnosed, the presence of five out of the nine following symptoms is required: depressed mood, lack of pleasure, difficulty sleeping and appetite, low energy levels, cognitive changes, feelings of guilt and guilt and suicidal ideas.

So far, the exact pathophysiological mechanisms for benefiting from effective treatment are not known. Several mechanisms have been called into question, but the theories are still developing. Thus, dysfunctions on the hypothalamic–pituitary axis, neurotransmitters, vagus nerve, short-chain fatty acid metabolites, tryptophan, inflammatory factors and the brain–gut–microbiota axis have been investigated [9,10,11,12]. It has been proved that not only do stress-related disorders bring changes in the composition of the intestinal microbiota through the hypothalamic–pituitary–adrenal axis but also the intestine has an effect on the central nervous system through vagal stimulation, intestinal permeability and the release of inflammatory and anti-inflammatory compounds and changes in circulating agents in the blood [13]. Over the past two decades, the study of brain–gut–microbiota axis involvement in mental disorders has gained momentum [9,14]. In this context, the role of leaky gut syndrome has received more attention [15,16].

The intestinal barrier has the main role in the digestion and absorption of nutrients; secondary, it strictly controls the transport of antigens from the lumen into the submucosa, and it maintains the balance between tolerance and the immune response that causes inflammation. The integrity of the intestinal barrier can be affected by diet, dysbiosis of the intestinal microbiota or by other factors. These have the potential to cause immune activation by translocation of microbial antigens and metabolites.

Currently, serum zonulin, intestinal fatty acid-binding protein (IFABP/FABP2) and LBP, as well as faecal albumin, calprotectin and zonulin, have been analysed as intestinal permeability markers [17].

Calprotectin is a protein that binds calcium and zinc, initially isolated from leukocytes; it is useful in differentiating inflammatory bowel disease from irritable bowel syndrome. It plays a major role during inflammation and is considered an acute phase protein [18]. Normal values are between 10 and 50 or 60 (ug/g). Values above 200 (ug/g) may indicate an inflammatory disease, and values above 500–600 (ug/g) certainly indicate the existence of an intestinal disease [19].

Lipopolysaccharide-binding protein (LBP) is an essential lipid component of the external membrane of Gram-negative bacteria [20], which causes endotoxemia [21]. LBP is produced by hepatocytes, adipose tissue and intestinal cells and functions as an acute phase I protein and plays an important role in the immune response [22]. LBP is present in serum at concentrations of 5–15 mg/mL and increases 10 to 50 times during the acute phase reaction.

The barrier function is maintained by zonulin, known as the only physiological modulator of close intracellular junctions (TJs) [15,23,24]. Zonulin reversibly opens the TJ, acting on protease-activated receptor 2 located in the intracellular space of the junctional complex [25]. The median value of zonulin is 34 ng/mL [26].

Intestinal fatty acid-binding protein (I-FABP/FABP2) is one of the nine different FABPs identified, which belong to a superfamily of proteins that bind lipids. The main role is in intracellular transport and the regulation of fatty acid absorption [27]. When damage to the epithelial intestinal barrier occurs, I-FABP is released into the circulation and its plasma concentration increases [28]. The threshold value for I-FAPB is considered 2 ng/mL.

In this context, we aimed to investigate the role of intestinal permeability biomarkers in relation to depression in patients with inflammatory bowel disease (IBD) by (i) examining the relationship between calprotectin, zonulin, LBP, IFABP/FABP2 and depression and by (ii) evaluating the influence of depression on the quality of life.

## 2. Materials and Methods

### 2.1. Study Group

The initial group included 60 patients from the Gastroenterology Clinic of the Constanta County Clinical Hospital between April and June 2021. The inclusion criteria were a diagnosis of IBD, age 18 and above, knowledge of reading and writing and agreement to participate in the study. The exclusion criteria were denial of consent, severe depression or other severe mental disorders for which they were directed to the specialised psychiatric service.

After informed consent, patients were contacted by phone by a researcher to apply a sociodemographic questionnaire on depression and a questionnaire on quality of life. Included in the study were 30 patients who went through all the stages of the study.

### 2.2. Questionnaires Used

The Patient Health Questionnaire-9 (PHQ-9) [29] was applied to participants in order to assess the level of depression by which they were affected. This standard questionnaire contains nine items to be responded to based on the symptoms the patient had during the last 14 days. The scores are classified as follows: 1–4 minimal depression, 5–9 mild depression, 10–14 moderate depression, 15–19 moderately severe depression and 20–27 severe depression

Similarly, we applied a questionnaire for quality of life (EQ-5D) [30,31] to collect data regarding quality of life. The items comprised in this tool are the mobility of the patient, self-care, engagement in usual activities, pain or discomfort and anxiety or depression experienced. The items are answered on a five-level scale.

### 2.3. Sample Collection and Laboratory Analysis

Serological samples for zonulin, IFABP/FABP2, LBP and faecal samples for calprotectin were collected according to the protocol for the collection and transport of potentially infectious biological samples. Serum samples were collected by venous puncture, fasting and in the vacutainer with an activator cloth. Faecal samples were collected from spontaneously emitted stool in containers without a carrier medium. The serum was tested for: zonulin with myBioSource kit, Inc. (San Diego, CA, USA). and IFABP/FABP2 and LBP with Wuhan Fine Biotech Co., Ltd. Kit (Wuhan, China).

The samples were tested for calprotectin with the EUROIMMUN Medizinische Labordiagnostika AG kit. All tests were performed by enzyme-related immunosorbent test (ELISA) on an ADALTIS Analyzer GEN-4 and Victor X4 according to the instructions of the kit manufacturer. The range of normal values was defined by the manufacturer of the kit only for calprotectin as follows: ≤50.0 mcg/g as normal, 50.1–120.0 mcg/g as the limit value and ≥120.1 mcg/g as positive for intestinal disease. For zonulin, the manufacturer of the reaction kit did not specify any lower detection limits.

The lower limit of detection for the IFABP/FABP2 kit was 0.156 ng/mL, 3.125 ng/mL for LBP and 3.125 ng/mL for calprotectin. For zonulin, I-FABP/FABP2 and LBS/LBP, the kit manufacturer did not communicate a range of normal values, and values used by other authors in their studies were used.

The serum was extracted from each blood sample by centrifugation at 3000× *g* for 30 min using a Thermo Scientific SL16R centrifuge. All the sera obtained were kept frozen at −200 °C until the analyses were carried out. For IFABP/FABP2 and LBP, the samples were diluted to 1/2 with a sample dilution buffer prior to testing. Stool samples were kept at −20 °C until the day of testing. Stool sample extraction was performed using a 980 iu extraction buffer along with 20 mg of stool, stirred for 30 s and then centrifuged at 2000× *g* for 10 min using the Thermo Scientific SL16R centrifuge. The dilution of the samples was then performed with a 980 iu of sample buffer and a 20 iu of extraction supernatant.

For the four markers, microplates with 96 wells coated with capture antibodies were used: anti-calprotectin antibodies, anti-IFABP/FABP2 antibodies, anti-LBP antibodies or anti-zonulin antibodies. The principles of the test were the same based on enzyme-related immunological testing technology (ELISA). Standard samples, test samples and biotin-conjugated detection antibodies were later added to the wells. TMB substrates (3,3′,5,5′—tetramethylbenzidine) were used to visualize the enzymatic reaction HRP (horseradish peroxidase). TMB was catalysed by HRP to produce a blue product that changed to yellow after the addition of an acid stop solution. The density of the yellow spot was proportional to the amount of sample captured in the plate. The colour intensity was measured at 450 nm in the microplate reader, and then the target concentration was calculated.

### 2.4. Statistical Analysis

Statistical analysis was performed using IBM SPSS statistics software version 23 (Armonk, NY, USA: IBM Corp.). Data are presented as median and IQR (interquartile range P75–P25) for continuous variables in case of skewed distributions or as percentages for categorical variables. Since the Shapiro–Wilk test of normality indicated, in the case of the continuous variables analysed, statistical values of the test whose probability (*p*) was lower than the level of significance α= 0.05, the condition of normality was not met, except for the variables LBP and PHQ 9, for which *p* > α = 0.05. We considered that a nonparametric approach to the study of the correlation between the variables in question, with the presentation of Spearman’s rho correlation coefficient (ρ) and the probability (*p*) associated with it, was more appropriate. The significance level α was set at 0.05.

## 3. Results

Thirty participants were involved in the study, presenting a balanced gender ratio. Of these, 40% of the participants were diagnosed with Crohn’s disease (CD) and 60% with ulcerative colitis (UC). There was an increased frequency of cases in the age category 30–40 years, 40–50 years and 50–60 years (Table 1).

### 3.1. Socioeconomic Conditions of the Studied Group

Considering civil status, a number of 20 (66.6%) married people participated in this study, and 10 (33.3%) were unmarried participants. Most of the participants (76.6%) lived in an urban area, and less than a quarter (23.3%) lived in a rural area. The educational level showed that 56.6 % graduated from university, while 20% graduated from high school and 23.3 % had other forms of education. Of the participants, 56.6% were full-time or part-time employees, and 43.3% of the participants benefited from various forms of social support (Table 1).

A high per cent (70%) of the participants found it difficult to comply with the recommended treatment, and only 30% of the patients easily followed the recommended diet. Almost half of the participants (46.6 %) performed minimal or no physical activity at all, while the rest (53.3 %) were engaged in regular physical activity for at least 30 min per day.

Concerning consumption habits, 26.6% of the participants declared themselves as smokers and 73.3% as nonsmokers; 33.3% of the participants stated they did not consume alcohol, while more than half (66.6%) stated they consumed alcohol (Table 1).

### 3.2. Central Tendency and Dispersion in Patients’ Health Score, Quality of Life and Biomarkers

For the PHQ-9 score, the median value was 6.50, with a maximum value of 16 and a minimum value of 0 (Table 1). A total of two (7%) participants had moderate–severe depression, five (17%) participants had moderate depression and 10 (33%) participants had mild depression. For 13 (43%) participants, depression was absent or minimal.

The median EQ-5D score was 80.00, with a maximum value of 100 and a minimum value of 45 (Table 2). Quality of life was assessed between the intervals 40 and 49 by one patient (3.33%), between 50 and –59 by three patients (10%) and between 60 and 69 by only one patient (3.33%) again, while for the interval 70–79 there were eight patients (26.67) that fitted, for 80–89 there were six patients (20%) and for 90–99 there were eight patients (26.67%). Only three patients (10%) assessed their quality of life to a score of 100.

For calprotectin, the median value was 149.64 ug/g, with a maximum value of 330.21 and a minimum value of 2.80 ug/g.

The LBP median value was 42.00 ng/mL, with a maximum value of 49.41 and a minimum value of 35.14 ng/mL.

In zonulin, the median value was 32.94 ng/mL, with a maximum value of 37.84 and a minimum value of 19.23 ng/mL.

In IFABP/FABP2, the median value was 0.87 ng/mL, with a maximum value of 3.00 and a minimum value of 0.35 ng/mL (Table 2).

### 3.3. Correlations between Patients’ Health Score and Biomarkers

A Spearman’s rank-order correlation was performed to determine the relationship between PHQ-9 score and calprotectin (ug/g), LBP (ng/mL), zonulin (ng/mL), IFABP/FABP2 (ng/mL) and EQ-5D.

There was a moderate, positive correlation between PHQ-9 score and calprotectin (ug/g) that was statistically significant (*r_s_*(30) = 0.416, *p* = 0.022, Table 3, Figure 1a), and a moderate, positive correlation between PHQ-9 score and LBP (ng/mL) that was also statistically significant (*r_s_*(30) = 0.398, *p* = 0.029, Table 3, Figure 1b).

Between PHQ-9 score and EQ-5D, we identified a weak, negative correlation that was statistically significant (*r_s_*(30) = −0.372, *p* = 0.043). No statistically significant correlation was found between PHQ-9 score and zonulin (ng/mL) (*r*_s_(30) = 0.016, *p* = 0.934) or between PHQ-9 score and IFABP/FABP2 (ng/mL) (*r_s_*(30) = −0.059, *p* = 0.755).

## 4. Discussion

The current study joins the literature that shows that depression is a frequent comorbidity of IBD and that common pathophysiological mechanisms can coexist [32]. The analysed data showed that more than half of the participants had depression with varying degrees of severity. In their recent study, Santosa and Galindo also showed an increased prevalence of anxiety and depression in patients with inflammatory bowel disease [33]. Among the pathophysiological mechanisms involved, the way the microbiota affects the signalling of the gut–brain axis is still under research. Alam showed that the recently introduced model of the “microbiota–gut–brain axis” has opened a new window for understanding the pathogenesis of neuropsychiatric syndromes, especially depression [33].

The hypothesis supported by Ait-Belgnaoui suggests that the microbiota is associated with microglial function, behaviour, affect, motivation and cognitive functions in animals, as well as in individuals with or without psychiatric diseases [34].

Moulton believed that inflammation provides a promising common origin for both depressive symptoms and the poor evolution of IBD [35]. Zonulin received considerable attention for its potential involvement in the pathogenesis of gastrointestinal disease and the possibility of being a biomarker of intestinal barrier dysfunction [22]. Sturgeron proposed, as an essential step in initiating the inflammatory process, the loss of intestinal barrier function by increasing zonulin [15], but our results do not show a significant correlation between zonulin levels and depression.

Similarly, Maget et al. found no significant difference between serum levels of zonulin in euthymic individuals and those with unipolar depression or depression in bipolar disorder. In addition, there was no significant correlation between the severity of depressive symptoms and serum levels of zonulin [36,37]. On the other hand, Kılıç found that zonulin is increased in patients with bipolar disorder and that this finding could contribute to the role of intestinal permeability or the blood–brain barrier in the pathogenesis of bipolar disorder [38]. Other studies have shown a correlation between zonulin and other mental disorders, e.g., mean levels of zonulin appear to be higher in children diagnosed with mental disorders compared with the control groups [39].

Wang [40] showed that an increase in zonulin is a significant factor in reducing the Mini-Mental State Examination (MMSE) score in mild cognitive deficiency and Alzheimer’s disease. Moreover, Ohlsson found that zonulin and IFAB 2 were altered in patients with a recent suicide attempt and that the “leaky gut hypothesis” could help explain part of the association between inflammation and suicidal behaviour [28]. Further investigations are needed [39].

In this study, we found that for calprotectin, the median value was above the normal range and that the value positively correlates with the PHQ 9 score. We believe that the inflammatory process hypothesis in depression can be confirmed. Similarly, Liśkiewicz found that there is a positive correlation with changes in faecal calprotectin during hospitalisation in patients with major depressive disorder and that intestinal integrity and markers of inflammation were associated with the response to treatment and with the severity of symptoms [41].

A large amount of LPS occurs through intestinal microbiota dysbiosis, stimulates LBP and then causes a proinflammatory response [42,43,44]. Köhler showed that endotoxin levels in the blood might be a contributing factor in the association between depressive symptoms and altered immune responses [45]. Depressive symptoms are often linked to an inflammatory response and increased inflammation, although these associations are not always consistent. In the studied group, we identified a positive correlation between LBP and depression. The compact range of values obtained in the case of LBP shows a homogeneous level of endotoxin stimulation (LPS) in the patients’ serum.

In men, higher depressive symptoms have been linked to increased ex vivo inflammatory responses to lipopolysaccharides (LPS), while in women, accentuated depressive symptoms have been linked to an attenuated inflammatory response [46].

In the IFABP/FABP2 studied group, the median value was situated in between the normal range, but maximal values exceeded the threshold. However, the correlation with depression was nonexistent.

This finding comes in contradiction with the results of Liśkiewicz, who found that there is a negative correlation between IFABP/FABP2 in plasma and major depression [41].

In the present study, the correlations between zonulin and IFAB 2 and depression are statistically insufficient. On the other hand, we identified two positive correlations, i.e., between depression levels assessed using the PHQ 9 questionnaire and calprotectin and LPS. Stevens et al. [47] found that zonulin and FABP2 were each significantly elevated in the depression versus nondepressive control groups. Anxiety and depressive disorders have been associated with intestinal dysbiosis and the growth of molecules of intestinal epithelial integrity in the blood of asymptomatic subjects with gastrointestinal physical suffering. These findings highlight the fact that the gut can be considered a new target for managing depression, especially in physically asymptomatic people with gastrointestinal disorders.

In addition, in the studied group, a negative correlation was identified between depression and quality of life, which confirms our secondary objective.

The current study has some limitations that should be addressed: it was an observational study; the sample size was small, limiting the generalisation of these results to a larger population; the patients with severe depression or other severe mental disorders were directed to a specialised psychiatric service for their own protection; we included an inflammatory marker of disease activity, calprotectin, but we did not use endoscopy for disease activity as an objective marker, which would have made it difficult to include some patients in the study; and we included patients with varying degrees of disease activity. Related variables that were analysed were smoking, alcohol consumption and type of treatment followed.

The inconsistency of the results regarding the relationship between intestinal biomarkers and depression may also be related to the fact that the methods of analysis are different among various studies. The type of collected samples can differ, i.e., either serum or faeces. In addition, the type of analysis can vary: there are tests that analyse interactions of the antibodies zonulin–antigen zonulin (immunosorbent) and a colourimetric HRP detection system to detect zonulinic antigen in the samples used in the present study or to identify serum levels of zonulin/preHP2. For these, further studies are needed to establish the primary target proteins in the family of zonulinic proteins [48] or structurally similar proteins detected by the available ELISA [49]. New and specific detection methods and assays for zonulin/preHP2 are urgently needed to address the usefulness of zonulin as a biomarker for intestinal permeability [50].

## 5. Conclusions

This is the first study in Romania, Eastern Europe, to include the set of biomarkers of intestinal permeability, i.e., calprotectin, zonulin, LBP and IFABP/FABP2, in the study of depression in patients with IBD for analysing whether intestinal permeability syndrome is correlated with depression. Our results highlighted a correlation between depression and calprotectin and LBP, which contributes another step to the rapid identification of biomarkers and can indicate the existence of depression.

Mental disorder identification can benefit from the model of the “microbiota–gut–brain axis”. Inflammation provides a common pathway with the potential for depression as well as IBD.

Depression and quality of life in patients with IBD are correlated, which urgently accelerates the need for access to appropriate treatments. The involvement of calprotectin, zonulin, LBP and IFABP/FABP2 in the study of mental disorders remains an open matter, given that inflammatory processes are known to be involved in the aetiology of affective disorders.

## Figures and Tables

**Figure 1 jcm-11-05121-f001:**
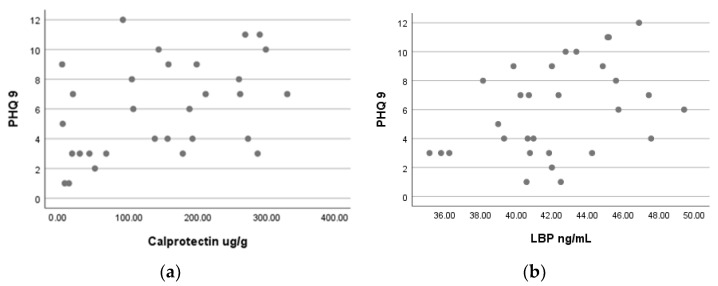
Spearman’s correlations between patients’ health score and biomarkers (**a**) calprotectin and (**b**) lipopolysaccharide-binding protein.

**Table 1 jcm-11-05121-t001:** Socioeconomic characterisation of the studied patients.

Parameter	Response	*n*	Percent (%)
Age (years) *	20–30	5	16.67
	30–40	6	20
	40–50	8	26.67
	50–60	7	23.33
	60–70	2	6.67
	70–80	2	6.67
Sex	Masculine	15	50
	Feminine	15	50
Origin	Urban	23	76.67
	Rural	7	23.33
Marital status	Single	10	33.33
	Married	20	66.67
Education level	Secondary school	6	20
	Faculty	17	56.67
	Others	7	23.33
Occupational status	Full-time employee	15	50
	Part-time employee	2	6.67
	Pensioner	7	23.33
	With disabilities	2	6.67
	Others	4	13.33
Diagnostic	Crohn’s disease	12	40
	Ulcerative colitis	18	60
Smoking status	Smoker	8	26.67
	Former smoker	12	40
	Nonsmoker	10	33.33
Alcohol consumption	Of	10	33.33
	Right away	20	66.67
Physical activity **	6–7x/W	11	36.67
	3–4x/S	5	16.67
	1–2x/S	8	26.67
	Not at all	6	20
Compliance with the diet	Easy	9	30
	Difficult	21	70

* Age, 30 includes years completed; ** 30 min physical activity/day for 1 week.

**Table 2 jcm-11-05121-t002:** Measures of central tendency and dispersion for the numerical variables included in the study.

Variable	Median	Min	Max	Range	IQR
PHQ-9 *	6.50	0.00	16.00	16.00	6.00
EQ-5D	80.00	45.00	100.00	55.00	20.00
Calprotectin (ug/g)	149.64	2.80	330.21	327.41	221.42
LBP (ng/mL)	42.00	35.14	49.41	14.27	4.99
Zonulin (ng/mL)	32.94	19.23	37.84	18.61	3.36
IFABP/FABP2 (ng/mL)	0.87	0.35	3.00	2.65	0.65

* PHQ-9, Patient Health Questionnaire; EQ-5D, questionnaire for quality of life; LBP, lipopolysaccharide-binding protein; I-FABP/FABP2, intestinal fatty acid-binding protein.

**Table 3 jcm-11-05121-t003:** Spearman’s correlation between patients’ health score and biomarkers and patients’ quality of life score.

	*n*	rho	*p* (2-Tailed)
Calprotectin (ug/g)	30	0.416	0.022
LBP * (ng/mL)	30	0.398	0.029
Zonulin (ng/mL)	30	0.016	0.934
IFABP/FABP2 (ng/mL)	30	−0.059	0.755
EQ-5D	30	−0.372	0.043

* LBP, lipopolysaccharide-binding protein; I-FABP/FABP2, intestinal fatty acid-binding protein; EQ-5D, questionnaire for quality of life.

## Data Availability

Not applicable.

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
