# Peer review of "Intestinal Permeability and Depression in Patients with Inflammatory Bowel Disease"

_jcm, 2022, doi:10.3390/jcm11175121_

Round 1
Reviewer 1 Report
The authors included 30 patients with inflammatory bowel disease, evaluated their depression severity with PHQ, and analyzed some gut permeability markers, such as serum levels of zonulin, calprotectin, lipopolysaccharide binding protein (LBP), and intestinal fatty acid binding protein (iFABP). They further evaluated the correlations between depression severity and these biomarkers. Some comments on the manuscript are listed below:
1. Because there is no control group, the authors had better to cite the references of normal range of these biomarkers in the manuscript.
2. The study excluded patients with severe depression from this study. So the depression severity is limited to a narrow range. PHQ scores range from 0-27. The study population’s PHQ range from 1-12, Narrow range of depression severity. In addition, no definition of depression severity by PHQ scores was mentioned in the manuscript.
3. Normality test should be put in statistical analysis, but not in the results.
4. The manuscript should reorganize.
5. Too many errors were found in the whole manuscript. Please check the manuscript in details. Some examples are listed in the following points 6-8.
6. I do not understand the first row of Table 1. Percentage of Marital status in Table 1 is wrong.
7. Minimal and maximal values of PHQ are inconsistent in table 1 and text.
8. The title of table 4 is quite strange. It should be the correlation between PHQ and biomarkers.
9. Results in text almost repeat numbers in Tables. For example, 3.1 section repeats Table 1.
10. Many variables show median in Tables, but average value in text. It makes confusing.
11. Results of normality test can be described in text, but not in one Table.
12. Discussion should focus on results. The first 3 paragraphs look like introduction, but no discussion.
13. Discussion should focus on depression, but not too many other mental disorders.
14. The authors can discuss or raise some possibilities to explain many inconsistent results about gut permeability and depression, but not just present inconsistent studies.
15. Since gut permeability is supposedly correlated with inflammation, the study did not evaluate the correlations between markers of gut permeability and inflammation, such as CRP, IL-6, etc.
Author Response
Dear reviewer,
We highly appreciated the constructive comments received that helped us improve our manuscript. We take the opportunity to thank you for the effort!
As suggested, we reviewed the whole manuscript in order to correct the existing errors, improve the structure of the paper and offer a stronger references list of relevant studies linked to our topic.
Please find below the point by point answers to the comments received.
--------------------------------------------------------------------------------------------------------
The authors included 30 patients with inflammatory bowel disease, evaluated their depression severity with PHQ, and analyzed some gut permeability markers, such as serum levels of zonulin, calprotectin, lipopolysaccharide binding protein (LBP), and intestinal fatty acid binding protein (iFABP). They further evaluated the correlations between depression severity and these biomarkers. Some comments on the manuscript are listed below:
- Because there is no control group, the authors had better to cite the references of normal range of these biomarkers in the manuscript.
Thank you for this suggestion. We included the missing information regarding normality range.
- The study excluded patients with severe depression from this study. So the depression severity is limited to a narrow range. PHQ scores range from 0-27. The study population’s PHQ range from 1-12, Narrow range of depression severity. In addition, no definition of depression severity by PHQ scores was mentioned in the manuscript.
We corrected the error which was the upper limit limited to 12 and now it reflects the reality of 16. Even though we understand the concern regarding the narrow range. Unfortunately it was not possible to perform further investigation in patients with severe depression for their own safety.
- Normality test should be put in statistical analysis, but not in the results.
Thank you for the comments. We made the modifications.
- The manuscript should reorganize.
We took into consideration your comment and made some adjustments in the manuscript concerning the structure.
- Too many errors were found in the whole manuscript. Please check the manuscript in details. Some examples are listed in the following points 6-8.
Thank you for your observations. We had reviewed the whole manuscript in order to correct these errors.
- I do not understand the first row of Table 1. Percentage of Marital status in Table 1 is wrong.
Thank you for the comments. We made the modifications.
- Minimal and maximal values of PHQ are inconsistent in table 1 and text.
We checked the text and solved this issue.
- The title of table 4 is quite strange. It should be the correlation between PHQ and biomarkers.
Thank you for the comments. We made the modifications.
- Results in text almost repeat numbers in Tables. For example, 3.1 section repeats Table 1.
We rewrote section 3.1 in order to contain less redundant information.
- Many variables show median in Tables, but average value in text. It makes confusing.
Because of the skewed distribution of the data we decided to stick with the median values in both text and tables.
- Results of normality test can be described in text, but not in one Table.
We took into consideration your comment and deleted the table. The argument of selecting a non-parametric test after performing the normality test was kept in the methods section.
- Discussion should focus on results. The first 3 paragraphs look like introduction, but no discussion.
Thank you for this observation. We corrected this issue.
- Discussion should focus on depression, but not too many other mental disorders.
We took into consideration this aspect and we only kept relevant information for depression.
- 1 The authors can discuss or raise some possibilities to explain many inconsistent results about gut permeability and depression, but not just present inconsistent studies.
The inconsistency of the results regarding the relationship of intestinal biomarkers and depression may be related to the fact that this is a relatively new field of research and a small number of studies are conducted so far, on small groups, which do not allow the extrapolation of the results at the level of the general population.
Some studies do not have control group or objective markers for IBD disease activity such as CRP, endoscopy.
The results may be influenced by the heterogeneity of the patients' disease, the presence or absence of drug treatment, as well as by unanalyzed variables related to smoking, alcohol consumption and eating habits.
Also, the methods of analysis are different. Type of samples: in the case of zonulin either serum or faeces; type of analysis: there are tests that analyse interactions of antibodies zonulin-antigen zonulin (immunosorbent) and a colorimetric HRP detection system to detect zonulinic antigen in samples, used in present study or identify serum levels of zonulin/preHP2. For this, further studies are needed to establish the primary target proteins in the family of zonulinic proteins or structurally similar proteins detected by the available ELISA. New and specific detection methods and assays for zonulin/preHP2 are urgently needed to address the usefulness of zonulin as a biomarker for intestinal permeability.
- Since gut permeability is supposedly correlated with inflammation, the study did not evaluate the correlations between markers of gut permeability and inflammation, such as CRP, IL-6, etc.
This is one of the limitations of the present study. Thank you for the useful suggestion. We will take this into consideration in our further research.
Reviewer 2 Report
Minor Considerations: Acronyms such as DSMV 5 must be spelled out in full the first time it is mentioned in the text.
Methodology: Regarding the methodology, I consider the sample size close to reasonable. However, there is little detail about the questionnaire and the PHQ and EQ scores. They should be better transcribed in the methodology, or at least adequate referencing to obtain details of both methodologies. As for the serum dosage, it is in line with the usual methodology in the scientific environment.
Although the normality test was used, the authors did not inform which was the statistical test and post-test of choice. I suggest better detailing, as well as informing which test was used in the legend of graphs and tables in the results.
Results: In the description of the results, I notice that the data obtained from the EQ score was not described, in the same way as the data from the PHQ score was stratified, thus, I suggest standardization and better exposure of the results. I understand that your work was approached only as three parameters: PHQ, EQ and serum dosages, in this way, the few results available should be better explained and discussed.
In the discussion, the results of the relationship between calprotectin (paragraphs 6 and 7) and LBP (paragraphs 8 and 9) with depression, and therefore microbiota-brain-gut axis, are not consistent, and are supported by a vague exposition of few bibliographic references. . Furthermore, it also brings several paragraphs of zonulin into the discussion (paragraphs 3 to 5) that add little to its discussion, since the authors did not show a relationship between zonulin and their findings. In this way, the text informed above is placed in the article unnecessarily. A discussion based on consistent results and correlated with the relevant literature is necessary.
Finally, I conclude that an extensive and rigorous review of the points listed above must be prepared by the authors.
Author Response
Dear reviewer,
We highly appreciated the constructive comments received that helped us improve our manuscript. We take the opportunity to thank you for the effort!
As suggested, we reviewed the whole manuscript in order to correct the existing errors, improve the structure of the paper and offer a stronger references list of relevant studies linked to our topic.
Please find below the point by point answers to the comments received.
--------------------------------------------------------------------------------------------------------
Minor Considerations: Acronyms such as DSM V 5 must be spelled out in full the first time it is mentioned in the text.
We spelled out the acronym.
Methodology: Regarding the methodology, I consider the sample size close to reasonable. However, there is little detail about the questionnaire and the PHQ and EQ scores. They should be better transcribed in the methodology, or at least adequate referencing to obtain details of both methodologies. As for the serum dosage, it is in line with the usual methodology in the scientific environment.
Thank you for this observation! We made modifications in the methodology section in order to clarify and complete the information. A brief description and citation was included for the questionnaires.
Although the normality test was used, the authors did not inform which was the statistical test and post-test of choice. I suggest better detailing, as well as informing which test was used in the legend of graphs and tables in the results.
Now we have included a description of the statistical test in the methods section in order to inform the reader. First a Shapiro-Wilk Test was used in order to examine the distribution of our data. Since the condition of normality was not met for the majority of the variables we decided to continue with nonparametric analysis, and we selected Spearman's correlations for identifying whether or not a link between depression and biomarkers exist.
Results: In the description of the results, I notice that the data obtained from the EQ score was not described, in the same way as the data from the PHQ score was stratified, thus, I suggest standardization and better exposure of the results. I understand that your work was approached only as three parameters: PHQ, EQ and serum dosages, in this way, the few results available should be better explained and discussed.
Thank you for the comment. We rewrote the results accordingly.
In the discussion, the results of the relationship between calprotectin (paragraphs 6 and 7) and LBP (paragraphs 8 and 9) with depression, and therefore microbiota-brain-gut axis, are not consistent, and are supported by a vague exposition of few bibliographic references. Furthermore, it also brings several paragraphs of zonulin into the discussion (paragraphs 3 to 5) that add little to its discussion, since the authors did not show a relationship between zonulin and their findings. In this way, the text informed above is placed in the article unnecessarily. A discussion based on consistent results and correlated with the relevant literature is necessary.
Finally, I conclude that an extensive and rigorous review of the points listed above must be prepared by the authors.
We had reviewed the discussion section and excluded the unnecessary information, as recommended. We further tried to lead the discussion around the relevant findings of our manuscript.
Round 2
Reviewer 1 Report
The authors have responded well to my comments.
No more comments.
Reviewer 2 Report
After the appropriate adjustments to the points raised, and all questions resolved.